# Unsupervised Learning via Meta-Learning

**Kyle Hsu**[†]
University of Toronto
kyle.hsu@mail.utoronto.ca

**Sergey Levine, Chelsea Finn**
University of California, Berkeley
{svlevine,cbfinn}@eecs.berkeley.edu

## Abstract

A central goal of unsupervised learning is to acquire representations from unlabeled data or experience that can be used for more effective learning of downstream tasks from modest amounts of labeled data. Many prior unsupervised learning works aim to do so by developing proxy objectives based on reconstruction, disentanglement, prediction, and other metrics. Instead, we develop an unsupervised meta-learning method that explicitly optimizes for the ability to learn a variety of tasks from small amounts of data. To do so, we construct tasks from unlabeled data in an automatic way and run meta-learning over the constructed tasks. Surprisingly, we find that, when integrated with meta-learning, relatively simple task construction mechanisms, such as clustering embeddings, lead to good performance on a variety of downstream, human-specified tasks. Our experiments across four image datasets indicate that our unsupervised meta-learning approach acquires a learning algorithm without any labeled data that is applicable to a wide range of downstream classification tasks, improving upon the embedding learned by four prior unsupervised learning methods.

## 1 Introduction

Unsupervised learning is a fundamental, unsolved problem (Hastie et al., 2009) and has seen promising results in domains such as image recognition (Le et al., 2013) and natural language understanding (Ramachandran et al., 2017). A central use case of unsupervised learning methods is enabling better or more efficient learning of downstream tasks by training on top of unsupervised representations (Reed et al., 2014; Cheung et al., 2015; Chen et al., 2016) or fine-tuning a learned model (Erhan et al., 2010). However, since the downstream objective requires access to supervision, the objectives used for unsupervised learning are only a rough proxy for downstream performance. If a central goal of unsupervised learning is to learn *useful* representations, can we derive an unsupervised learning objective that explicitly takes into account how the representation will be used?

The use of unsupervised representations for downstream tasks is closely related to the objective of meta-learning techniques: finding a learning procedure that is more efficient and effective than learning from scratch. However, unlike unsupervised learning methods, meta-learning methods require large, labeled datasets and hand-specified task distributions. These dependencies are major obstacles to widespread use of these methods for few-shot classification.

To begin addressing these problems, we propose an unsupervised meta-learning method: one which aims to learn a learning procedure, without supervision, that is useful for solving a wide range of new, human-specified tasks. With only raw, unlabeled observations, our model's goal is to learn a useful prior such that, after meta-training, when presented with a modestly-sized dataset for a human-specified task, the model can transfer its prior experience to efficiently learn to perform the new task. If we can build such an algorithm, we can enable few-shot learning of new tasks without needing any labeled data nor any pre-defined tasks.

To perform unsupervised meta-learning, we need to automatically construct tasks from unlabeled data. We study several options for how this can be done. We find that a good task distribution should be diverse, but also not too difficult: naïve random approaches for task generation produce tasks that contain insufficient regularity to enable useful meta-learning. To that end, our method proposes tasks by first leveraging prior unsupervised learning algorithms to learn an embedding of the

---

[†]Work done as a visiting student researcher at the University of California, Berkeley.

input data, and then performing an overcomplete partitioning of the dataset to construct numerous categorizations of the data. We show how we can derive classification tasks from these categorizations for use with meta-learning algorithms. Surprisingly, even with simple mechanisms for partitioning the embedding space, such as $k$-means clustering, we find that meta-learning acquires priors that, when used to learn new, human-designed tasks, learn those tasks more effectively than methods that directly learn on the embedding. That is, the learning algorithm acquired through unsupervised meta-learning achieves better downstream performance than the original representation used to derive meta-training tasks, without introducing any additional assumptions or supervision. See Figure 1 for an illustration of the complete approach.

The core idea in this paper is that we can leverage unsupervised embeddings to propose tasks for a meta-learning algorithm, leading to an unsupervised meta-learning algorithm that is particularly effective as pre-training for human-specified downstream tasks. In the following sections, we formalize our problem assumptions and goal, which match those of unsupervised learning, and discuss several options for automatically deriving tasks from embeddings. We instantiate our method with two meta-learning algorithms and compare to prior state-of-the-art unsupervised learning methods. Across four image datasets (MNIST, Omniglot, miniImageNet, and CelebA), we find that our method consistently leads to effective downstream learning of a variety of human-specified tasks, including character recognition tasks, object classification tasks, and facial attribute discrimination tasks, without requiring any labels or hand-designed tasks during meta-learning and where key hyperparameters of our method are held constant across all domains. We show that, even though our unsupervised meta-learning algorithm trains for one-shot generalization, one instantiation of our approach performs well not only on few-shot learning, but also when learning downstream tasks with up to $50$ training examples per class. In fact, some of our results begin to approach the performance of fully-supervised meta-learning techniques trained with fully-specified task distributions.

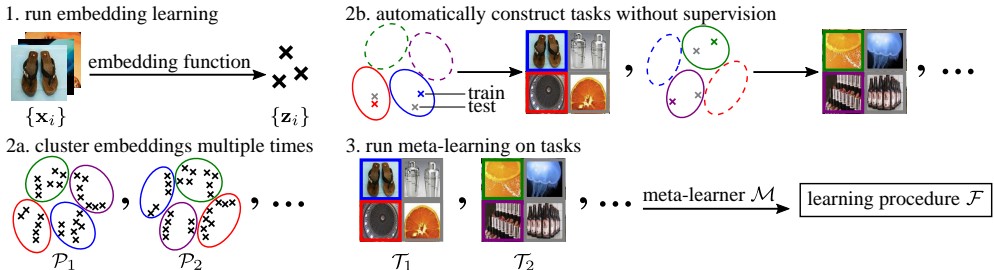

Figure 1: Illustration of the proposed unsupervised meta-learning procedure. Embeddings of raw observations are clustered with $k$-means to construct partitions, which give rise to classification tasks. Each task involves distinguishing between examples from $N = 2$ clusters, with $K_{\text{m-tr}} = 1$ example from each cluster being a training input. The meta-learner's aim is to produce a learning procedure that successfully solves these tasks.

## 2 UNSUPERVISED META-LEARNING

In this section, we describe our problem setting in relation to that of unsupervised and semi-supervised learning, provide necessary preliminaries, and present our approach.

### 2.1 PROBLEM STATEMENT

Our goal is to leverage unlabeled data for the efficient learning of a range of human-specified downstream tasks. We only assume access to an unlabeled dataset $\mathcal{D} = \{\mathbf{x}_i\}$ during meta-training. After learning from the unlabeled data, which we will refer to as unsupervised meta-training, we want to apply what was learned towards learning a variety of downstream, human-specified tasks from a modest amount of labeled data, potentially as few as a single example per class. These downstream tasks may, in general, have different underlying classes or attributes (in contrast to typical semi-supervised problem assumptions), but are assumed to have inputs from the same distribution as the one from which datapoints in $\mathcal{D}$ are drawn. Concretely, we assume that downstream tasks are $M$-way classification tasks, and that the goal is to learn an accurate classifier using $K$ labeled datapoints $(\mathbf{x}_k, \mathbf{y}_k)$ from each of the $M$ classes, where $K$ is relatively small (i.e. between 1 and 50).

The unsupervised meta-training phase aligns with the unsupervised learning problem in that it involves no access to information about the downstream tasks, other than the fact that they are $M$-way classification tasks, for variable $M$ upper-bounded by $N$. The upper bound $N$ is assumed to be known during unsupervised meta-training, but otherwise, the values of $M$ and $K$ are not known *a priori*. As a result, the unsupervised meta-training phase needs to acquire a sufficiently general prior for applicability to a range of classification tasks with variable quantities of data and classes. This problem definition is our prototype for a practical use-case in which a user would like to train an application-specific image classifier, but does not have an abundance of labeled data.

## 2.2 PRELIMINARIES

**Unsupervised embedding learning**. An unsupervised embedding learning algorithm $\mathcal{E}$ is a procedure that takes as input an unlabeled dataset $\mathcal{D} = \{\mathbf{x}_i\}$ and outputs a mapping from $\{\mathbf{x}_i\}$ to embeddings $\{\mathbf{z}_i\}$. These embedded points are typically lower-dimensional and arranged such that distances correspond to meaningful differences between inputs, in contrast to distances between the original inputs, such as image pixels, which are not meaningful measures of image similarity.

**Task**. An $M$-way $K$-shot classification task $\mathcal{T}$ consists of $K$ training datapoints and labels $\{(\mathbf{x}_k, \boldsymbol{\ell}_k)\}$ per class, which are used for learning a classifier, and $Q$ query datapoints and labels per class, on which the learned classifier is evaluated. That is, in a task there are $K + Q = R$ datapoints and labels for each of the $M$ classes.

**Meta-learning**. A supervised meta-learning algorithm $\mathcal{M}(\cdot)$ takes as input a set of supervised meta-training tasks $\{\mathcal{T}_t\}$. It produces a learning procedure $\mathcal{F}(\cdot)$, which, in turn, ingests the supervised training data of a task to produce a classifier $f(\cdot)$. The goal of $\mathcal{M}$ is to learn $\mathcal{F}$ such that, when faced with a meta-test time task $\mathcal{T}_{t'}$ held-out from $\{\mathcal{T}_t\}$, $\mathcal{F}$ can learn a $f_{t'}$ that accomplishes $\mathcal{T}_{t'}$. At a high level, the quintessential meta-learning strategy is to have $\mathcal{M}$ iterate over $\{\mathcal{T}_t\}$, cycling between applying the current form of $\mathcal{F}_t$ on training data from $\mathcal{T}_t$ to learn $f_t$, assessing its performance by calculating some meta-loss $\mathcal{L}$ on held-out data from the task, and optimizing $\mathcal{L}$ to improve the learning procedure.

We build upon two meta-learning algorithms: model agnostic meta-learning (MAML) (Finn et al., 2017) and prototypical networks (ProtoNets) (Snell et al., 2017). MAML aims to learn the initial parameters of a deep network such that one or a few gradient steps leads to effective generalization; it specifies $\mathcal{F}$ as gradient descent starting from the meta-learned parameters. ProtoNets aim to meta-learn a representation in which a class is effectively identified by its prototype, defined to be the mean of the class' training examples in the meta-learned space; $\mathcal{F}$ is the computation of these class prototypes, and $f$ is a linear classifier that predicts the class whose prototype is closest in Euclidean distance to the query's representation.

**Task generation for meta-learning**. We briefly summarize how tasks are typically generated from labeled datasets $\{(\mathbf{x}_i, \mathbf{y}_i)\}$ for supervised meta-learning, as introduced by Santoro et al. (2016). For simplicity, consider the case where the labels are discrete scalar values $y_i$. To construct an $N$-way classification task $\mathcal{T}$ (assuming $N$ is not greater than the number of unique $y_i$), we can sample $N$ classes, sample $R$ datapoints $\{\mathbf{x}_r\}_n$ for each of the $N$ classes, and sample a permutation of $N$ distinct one-hot vectors $(\boldsymbol{\ell}_n)$ to serve as task-specific labels of the $N$ sampled classes. The task is then defined as $\mathcal{T} = \{(\mathbf{x}_{n,r}, \boldsymbol{\ell}_n) \mid \mathbf{x}_{n,r} \in \{\mathbf{x}_r\}_n\}$. Of course, this procedure is only possible with labeled data; in the next section, we discuss how we can construct tasks without ground-truth labels.

## 2.3 UNSUPERVISED META-LEARNING WITH AUTOMATICALLY CONSTRUCTED TASKS

We approach our problem from a meta-learning perspective, framing the problem as the acquisition, from unlabeled data, of an efficient learning procedure that is transferable to human-designed tasks. In particular, we aim to construct classification tasks from the unlabeled data and then learn how to efficiently learn these tasks. If such tasks are adequately *diverse* and *structured*, then meta-learning these tasks should enable fast learning of new, human-provided tasks. A key question, then, is how to automatically construct such tasks from unlabeled data $\mathcal{D} = \{\mathbf{x}_i\}$. Notice that in the supervised meta-learning task generation procedure detailed in Section 2.2, the labels $y_i$ induce a partition $\mathcal{P} = \{\mathcal{C}_c\}$ over $\{\mathbf{x}_i\}$ by assigning all datapoints with label $y_c$ to subset $\mathcal{C}_c$. Once a partition is obtained, task generation is simple; we can reduce the problem of constructing tasks to that of

constructing a partition over $\{\mathbf{x}_i\}$. All that's left is to find a principled alternative to human labels for defining the partition.

A naïve approach is to randomly partition the data $\mathcal{D}$. While such a scheme introduces diverse tasks, there is no structure; that is, there is no consistency between a task's training data and query data, and hence nothing to be learned during each task, let alone across tasks. As seen in Table 3, providing a meta-learner with purely random tasks results in failed meta-learning.

To construct tasks with structure that resembles that of human-specified labels, we need to group datapoints into consistent and distinct subsets based on salient features. With this motivation in mind, we propose to use $k$-means clustering. Consider the partition $\mathcal{P} = \{\mathcal{C}_c\}$ learned by $k$-means as a simplification of a Gaussian mixture model $p(\mathbf{x}|c)p(c)$. If the clusters can recover a semblance of the true class-conditional generative distributions $p(\mathbf{x}|c)$, creating tasks based on treating these clusters as classes should result in useful unsupervised meta-training. However, the result of $k$-means is critically dependent on the metric space on which its objective is defined. Clustering in pixel-space is unappealing for two reasons: (1) distance in pixel-space correlates poorly with semantic meaning, and (2) the high dimensionality of raw images renders clustering difficult in practice. We empirically show in Table 3 that meta-learning with tasks defined by pixel-space clusters, with preprocessing as directed by Coates & Ng (2012), also fails.

We are now motivated to cluster in spaces in which common distance functions correlate to semantic meaning. However, we must satisfy the constraints of our problem statement in the process of learning such spaces. To these ends, we use state-of-the-art unsupervised learning methods to produce useful embedding spaces. For qualitative evidence in the unsupervised learning literature that such embedding spaces exhibit semantic meaning, see Cheung et al. (2015); Bojanowski & Joulin (2017); Donahue et al. (2017). We note that while a given embedding space may not be directly suitable for highly-efficient learning of new tasks (which would require the embedding space to be precisely aligned or adaptable to the classes of those tasks), we can still leverage it for the construction of structured tasks, a process for which requirements are less strict.

Thus, we first run an out-of-the-box unsupervised embedding learning algorithm $\mathcal{E}$ on $\mathcal{D}$, then map the data $\{\mathbf{x}_i\}$ into the embedding space $\mathcal{Z}$, producing $\{\mathbf{z}_i\}$. To produce a diverse task set, we generate $P$ partitions $\{\mathcal{P}_p\}$ by running clustering $P$ times, applying random scaling to the dimensions of $\mathcal{Z}$ to induce a different metric, represented by diagonal matrix $\mathbf{A}$, for each run of clustering. With $\boldsymbol{\mu}_c$ denoting the learned centroid of cluster $\mathcal{C}_c$, a single run of clustering can be summarized with

$$\mathcal{P}, \{\boldsymbol{\mu}_c\} = \underset{\{\mathcal{C}_c\}, \{\boldsymbol{\mu}_c\}}{\arg\min} \sum_{c=1}^{k} \sum_{\mathbf{z} \in \mathcal{C}_c} \|\mathbf{z} - \boldsymbol{\mu}_c\|_{\mathbf{A}}^2 \tag{1}$$

We derive tasks for meta-learning from the partitions using the procedure detailed in Section 2.2, except we begin the construction of each task by sampling a partition from the uniform distribution $\mathcal{U}(\mathcal{P})$, and for $\mathbf{x}_i \in \mathcal{C}_c$, specify $y_i = c$. To avoid imbalanced clusters dominating the meta-training tasks, we opt not to sample from $p(c) \propto |\mathcal{C}_c|$, but instead sample $N$ clusters uniformly without replacement for each task. We note that Caron et al. (2018) are similarly motivated in their design decision of sampling data from a uniform distribution over clusters.

With the partitions being constructed over $\{\mathbf{z}_i\}$, we have one more design decision to make: should we perform meta-learning on embeddings or images? We consider that, to successfully solve new tasks at meta-test time, a learning procedure $\mathcal{F}$ that takes embeddings as input would depend on the embedding function's ability to generalize to out-of-distribution observations. On the other hand, by meta-learning on images, $\mathcal{F}$ can separately adapt $f$ to each evaluation task from the rawest level of representation. Thus, we choose to meta-learn on images.

We call our method clustering to automatically construct tasks for unsupervised meta-learning (CACTUs). We detail the task construction algorithm in Algorithm 1, and provide an illustration of the complete unsupervised meta-learning approach for classification in Figure 1.

## 3 RELATED WORK

The method we propose aims to address the unsupervised learning problem (Hastie et al., 2009; Le et al., 2013), namely acquiring a transferable learning procedure without labels. We show that our

---

**Algorithm 1** CACTUs for classification

---
1: **procedure** CACTUS($\mathcal{E}, \mathcal{D}, P, k, T, N, K_{\text{m-tr}}, Q$)
2:  Run embedding learning algorithm $\mathcal{E}$ on $\mathcal{D}$ and produce embeddings $\{\mathbf{z}_i\}$ from observations $\{\mathbf{x}_i\}$.
3:  Run $k$-means on $\{\mathbf{z}_i\}$ $P$ times (with random scaling) to generate a set of partitions $\{\mathcal{P}_p = \{\mathcal{C}_c\}_p\}$.
4:  **for** $t$ from 1 to the number of desired tasks $T$ **do**
5:      Sample a partition $\mathcal{P}$ uniformly at random from the set of partitions $\{\mathcal{P}_p\}$.
6:      Sample a cluster $\mathcal{C}_n$ uniformly without replacement from $\mathcal{P}$ for each of the $N$ classes desired for a task.
7:      Sample an embedding $\mathbf{z}_r$ without replacement from $\mathcal{C}_n$ for each of the $R = K_{\text{m-tr}} + Q$ training and query examples desired for each class, and record the corresponding datapoint $\mathbf{x}_{n,r}$.
8:      Sample a permutation $(\boldsymbol{\ell}_n)$ of $N$ one-hot labels.
9:      Construct $\mathcal{T}_t = \{(\mathbf{x}_{n,r}, \boldsymbol{\ell}_n)\}$.
10:  **return** $\{\mathcal{T}_t\}$

---

method is complementary to a number of unsupervised learning methods, including ACAI (Berthelot et al., 2018), BiGAN (Donahue et al., 2017; Dumoulin et al., 2017), DeepCluster (Caron et al., 2018), and InfoGAN (Chen et al., 2016): we leverage these prior methods to learn embeddings used for constructing meta-learning tasks, and demonstrate that our method learns a more useful representation than the embeddings. The ability to use what was learned during unsupervised pre-training to better or more efficiently learn a variety of downstream tasks is arguably one of the most practical applications of unsupervised learning methods, which has a long history in neural network training (Hinton et al., 2006; Bengio et al., 2007; Ranzato et al., 2006; Vincent et al., 2008; Erhan et al., 2010). Unsupervised pre-training has demonstrated success in a number of domains, including speech recognition (Yu et al., 2010), image classification (Zhang et al., 2017), machine translation (Ramachandran et al., 2017), and text classification (Dai & Le, 2015; Howard & Ruder, 2018; Radford et al., 2018). Our approach, unsupervised meta-learning, can be viewed as an unsupervised learning algorithm that explicitly optimizes for few-shot transferability. As a result, we can expect it to better learn human-specified downstream tasks, compared to unsupervised learning methods that optimize for other metrics, such as reconstruction (Vincent et al., 2010; Higgins et al., 2017), fidelity of constructed images (Radford et al., 2016; Salimans et al., 2016; Donahue et al., 2017; Dumoulin et al., 2017), representation interpolation (Berthelot et al., 2018), disentanglement (Bengio et al., 2013; Reed et al., 2014; Cheung et al., 2015; Chen et al., 2016; Mathieu et al., 2016; Denton & Birodkar, 2017), and clustering (Coates & Ng, 2012; Krähenbühl et al., 2016; Bojanowski & Joulin, 2017; Caron et al., 2018). We empirically evaluate this hypothesis in the next section. In contrast to many previous evaluations of unsupervised pre-training, we focus on settings in which only a small amount of data for the downstream tasks is available, since this is where the unlabeled data can be maximally useful.

Unsupervised pre-training followed by supervised learning can be viewed as a special case of the semi-supervised learning problem (Zhu, 2011; Kingma et al., 2014; Rasmus et al., 2015; Oliver et al., 2018). However, in contrast to our problem statement, semi-supervised learning methods assume that a significant proportion of the unlabeled data, if not all of it, shares underlying labels with the labeled data. Additionally, our approach and other unsupervised learning methods are well-suited for transferring their learned representation to many possible downstream tasks or labelings, whereas semi-supervised learning methods typically optimize for performance on a single task, with respect to a single labeling of the data.

Our method builds upon the ideas of meta-learning (Schmidhuber, 1987; Bengio et al., 1991; Naik & Mammone, 1992) and few-shot learning (Santoro et al., 2016; Vinyals et al., 2016; Ravi & Larochelle, 2017; Munkhdalai & Yu, 2017; Snell et al., 2017). We apply two meta-learning algorithms, model-agnostic meta-learning (Finn et al., 2017) and prototypical networks (Snell et al., 2017), to tasks constructed in an unsupervised manner. Similar to our problem setting, some prior works have aimed to learn an unsupervised learning procedure with supervised data (Garg & Kalai, 2017; Metz et al., 2018). Instead, we consider a problem setting that is entirely unsupervised, aiming to learn efficient learning algorithms using unlabeled datasets. Our problem setting is similar to that considered by Gupta et al. (2018), but we develop an approach that is suitable for supervised downstream tasks, rather than reinforcement learning problems, and demonstrate our algorithm on problems with high-dimensional visual observations.

## 4 EXPERIMENTS

We begin the experimental section by presenting our research questions and how our experiments are designed to address them. Links to code for the experiments can be found at `https://sites.google.com/view/unsupervised-via-meta`.

**Benefit of meta-learning**. Is there any significant benefit to doing meta-learning on tasks derived from embeddings, or is the embedding function already sufficient for downstream supervised learning of new tasks? To investigate this, we run MAML and ProtoNets on tasks generated via CACTUs (CACTUs-MAML, CACTUs-ProtoNets). We compare to five alternate algorithms, with four being supervised learning methods on top of the embedding function. i) Embedding $k_{nn}$-nearest neighbors first infers the embeddings of the downstream task images. For a query test image, it predicts the plurality vote of the labels of the $k_{nn}$ training images that are closest in the embedding space to the query's embedding. ii) Embedding linear classifier also begins by inferring the embeddings of the downstream task images. It then fits a linear classifier using the $NK$ training embeddings and labels, and predicts labels for the query embeddings using the classifier. iii) Embedding multilayer perceptron instead uses a network with one hidden layer of 128 units and tuned dropout (Srivastava et al., 2014). iv) To isolate the effect of meta-learning on images, we also compare to embedding cluster matching, i.e. directly using the meta-training clusters for classification by labeling clusters with a task's training data via plurality vote. If a query datapoint maps to an unlabeled cluster, the closest labeled cluster is used. v) As a baseline, we forgo any unsupervised pre-training and train a model with the MAML architecture from standard random network initialization via gradient descent separately for each evaluation task.

**Different embedding spaces**. Does CACTUs result in successful meta-learning for many distinct task-generating embeddings? To investigate this, we run unsupervised meta-learning using four embedding learning algorithms: ACAI (Berthelot et al., 2018), BiGAN (Donahue et al., 2017), DeepCluster (Caron et al., 2018), and InfoGAN (Chen et al., 2016). These four approaches collectively cover the following range of objectives and frameworks in the unsupervised learning literature: generative modeling, two-player games, reconstruction, representation interpolation, discriminative clustering, and information maximization. We describe these methods in more detail in Appendix A.

**Applicability to different tasks**. Can unsupervised meta-learning yield a good prior for a variety of task types? In other words, can unsupervised meta-learning yield a good representation for tasks that assess the ability to distinguish between features on different scales, or tasks with various amounts of supervision signal? To investigate this, we evaluate our procedure on tasks assessing recognition of character identity, object identity, and facial attributes. For this purpose we choose to use the existing Omniglot (Santoro et al., 2016) and miniImageNet (Ravi & Larochelle, 2017) datasets and few-shot classification tasks and, inspired by Finn et al. (2018), also construct a new few-shot classification benchmark based on the CelebA dataset and its binary attribute annotations. For miniImageNet, we consider both few-shot downstream tasks and tasks involving larger datasets (up to 50-shot). Specifics on the datasets and human-designed tasks are presented in Appendix B.

**Oracle**. How does the performance of our unsupervised meta-learning method compare to supervised meta-learning with a human-specified, near-optimal task distribution derived from a labeled dataset? To investigate this, we use labeled versions of the meta-training datasets to run MAML and ProtoNets as supervised meta-learning algorithms (Oracle-MAML, Oracle-ProtoNets). To facilitate fair comparison with the unsupervised variants, we control for the relevant hyperparameters.

**Task construction ablation**. How do the alternatives for constructing tasks from the embeddings compare? To investigate this, we run MAML on tasks constructed via clustering (CACTUs-MAML) and MAML on tasks constructed via random hyperplane slices of the embedding space with varying margin (Hyperplanes-MAML). The latter partitioning procedure is detailed in Appendix C. For the experiments where tasks are constructed via clustering, we also investigate the effect of sampling based on a single partition versus multiple partitions. We additionally experiment with tasks based on random assignments of images to "clusters" (Random-MAML) and tasks based on pixel-space clusters (Pixels CACTUs-MAML) with the Omniglot dataset.

To investigate the limitations of our method, we also consider an easier version of our problem statement where the data distributions at meta-training and meta-test time perfectly overlap, i.e. the images share a common set of underlying labels (Appendix D). Finally, we present results on miniImageNet after unsupervised meta-learning on most of ILSVRC 2012 (Appendix G).

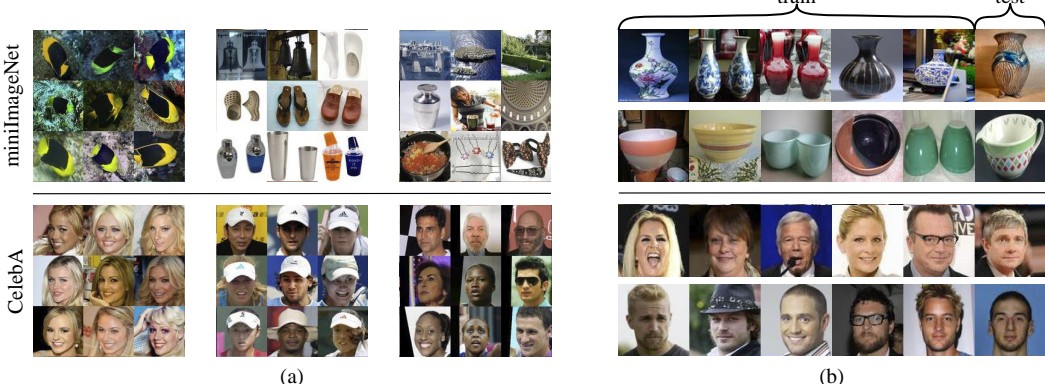

Figure 2: Examples of three DeepCluster-embedding cluster-based classes (a) and a 2-way 5-shot test task (b) for two datasets. (a) Some of the clusters correspond well to unseen labels (top left, bottom left). Others exhibit semantic meaning despite members not being grouped as such in the labeled version of the dataset (top middle: pair of objects, bottom middle: white hat). Still others are uninterpretable (top right) or are based on image artifacts (bottom right). (b) We evaluate unsupervised pre-training based on the ability to learn downstream, human-designed tasks with held-out images and underlying classes.

## 4.1 EXPERIMENTAL PROTOCOL SUMMARY

As discussed by Oliver et al. (2018), keeping proper experimental protocol is particularly important when evaluating unsupervised and semi-supervised learning algorithms. Our foremost concern is to avoid falsely embellishing the capabilities of our approach by overfitting to the specific datasets and task types that we consider. To this end, we adhere to two key principles. We do not perform any architecture engineering: we use architectures from prior work as-is, or lightly adapt them to our needs if necessary. We also keep hyperparameters related to the unsupervised meta-learning stage as constant as possible across all experiments, including the MAML and ProtoNets model architectures. Details on hyperparameters and architectures are presented in Appendix E. We assume knowledge of an upper bound on the number of classes $N$ present in each downstream meta-testing task for each dataset. However, regardless of the number of shots $K$, we do not assume knowledge of $K$ during unsupervised meta-learning. We use $N$-way 1-shot tasks during meta-training, but test on larger values of $K$ during meta-testing.

We partition each dataset into meta-training, meta-validation, and meta-testing splits. For Omniglot and miniImageNet, these splits contain disjoint sets of classes. For all algorithms, we run unsupervised pre-training on the unlabeled meta-training split and report performance on downstream tasks dictated by the *labeled* data of the meta-testing split, generated using the procedure from prior work recounted in Section 2.2. For the supervised meta-learning oracles, meta-training tasks are constructed in the same manner but from the dataset's meta-training split. See Figure 2 for illustrative examples of embedding-derived clusters and human-designed test tasks.

To facilitate analysis on meta-overfitting, we use the labels of the meta-validation split (instead of clustering embeddings) to construct tasks for meta-validation. However, because our aim is to perform meta-learning without supervision, we do not tune hyperparameters on this labeled data. We use a fixed number of meta-training iterations, since there is no suitable criterion for early stopping.

When we experiment with the embedding-plus-supervised-learning methods used as fair comparisons to unsupervised meta-learning, we err on the side of providing more supervision and data than technically allowed. Specifically, we separately tune the supervised learning hyperparameters for each dataset and each task difficulty on the labeled version of the meta-validation split. With Deep-Cluster embeddings, we also use the entire meta-testing split's statistics to perform dimensionality reduction (via PCA) and whitening, which is unfair as this shares information across tasks.

## 4.2 RESULTS

Our primary results are summarized in Tables 1 and 2. Task construction ablations are summarized in Tables 3 and 4.

**Benefit of meta-learning**. CACTUs-MAML consistently yields a learning procedure that results in more successful downstream task performance than all other unsupervised methods, including those that learn on top of the embedding that generated meta-training tasks for MAML. We find the same result for CACTUs-ProtoNets for 1-shot downstream tasks. However, as noted by Snell et al. (2017), ProtoNets perform best when meta-training shot and meta-testing shot are matched; this characteristic prevents ProtoNets from improving upon ACAI for 20-way 5-shot Omniglot and upon DeepCluster for 50-shot miniImageNet. We attribute the success of CACTUs-based meta-learning over the embedding-based methods to two factors: its practice in distinguishing between many distinct sets of clusters from modest amounts of signal, and the underlying classes of the meta-testing split data being out-of-distribution. In principle, the latter factor is solely responsible for the success over embedding cluster matching, since this algorithm can be viewed as a meta-learner on embeddings that trivially obtains perfect accuracy (via memorization) on the meta-training tasks. The same factor also helps explain why training from standard network initialization is, in general, competitive with directly using the task-generating embedding as a representation. On the other hand, the MNIST results (Table 7 in Appendix F) suggest that when the meta-training and meta-testing data distributions have perfect overlap and the embedding is well-suited enough that embedding cluster matching can already achieve high performance, CACTUs-MAML yields only a small benefit, as we would expect.

Table 1: Results of unsupervised learning on Omniglot images, averaged over 1000 downstream character recognition tasks. CACTUs experiments use $k = 500$ clusters for each of $P = 100$ partitions. Embedding cluster matching uses the same $k$. For complete results with confidence intervals, see Table 8 in Appendix F.

| Algorithm                          (way, shot) | (5, 1) | (5, 5) | (20, 1) | (20, 5) |
|-----------------------------------------------|--------|--------|---------|---------|
| Training from scratch                         | 52.50% | 74.78% | 24.91%  | 47.62%  |
| ACAI $k_{nn}$-nearest neighbors               | 57.46% | 81.16% | 39.73%  | 66.38%  |
| ACAI linear classifier                        | 61.08% | 81.82% | 43.20%  | 66.33%  |
| ACAI MLP with dropout                         | 51.95% | 77.20% | 30.65%  | 58.62%  |
| ACAI cluster matching                         | 54.94% | 71.09% | 32.19%  | 45.93%  |
| ACAI CACTUs-MAML (ours)                       | **68.84%** | **87.78%** | **48.09%** | **73.36%** |
| ACAI CACTUs-ProtoNets (ours)                  | **68.12%** | 83.58% | **47.75%** | 66.27%  |
| BiGAN $k_{nn}$-nearest neighbors              | 49.55% | 68.06% | 27.37%  | 46.70%  |
| BiGAN linear classifier                       | 48.28% | 68.72% | 27.80%  | 45.82%  |
| BiGAN MLP with dropout                        | 40.54% | 62.56% | 19.92%  | 40.71%  |
| BiGAN cluster matching                        | 43.96% | 58.62% | 21.54%  | 31.06%  |
| BiGAN CACTUs-MAML (ours)                      | 58.18% | 78.66% | 35.56%  | 58.62%  |
| BiGAN CACTUs-ProtoNets (ours)                 | 54.74% | 71.69% | 33.40%  | 50.62%  |
| Oracle-MAML (control)                         | 94.46% | 98.83% | 84.60%  | 96.29%  |
| Oracle-ProtoNets (control)                    | 98.35% | 99.58% | 95.31%  | 98.81%  |

**Different embedding spaces**. CACTUs is effective for a variety of embedding learning methods used for task generation. The performance of unsupervised meta-learning can largely be predicted by the performance of the embedding-based non-meta-learning methods. For example, the ACAI embedding does well with Omniglot, leading to the best unsupervised results with ACAI CACTUs-MAML. Likewise, on miniImageNet, the best performing prior embedding (DeepCluster) also corresponds to the best performing unsupervised meta-learner (DeepCluster CACTUs-MAML).

**Applicability to different tasks**. CACTUs-MAML learns an effective prior for a variety of task types. This can be attributed to the application-agnostic task-generation process and the expressive power of MAML (Finn & Levine, 2018). We also observe that, despite all meta-learning models being trained for $N$-way 1-shot classification of unsupervised tasks, the models work well for a variety of $M$-way $K$-shot tasks, where $M \leq N$ and $K \leq 50$. As mentioned previously, the representation that CACTUs-ProtoNets learns is best suited for downstream tasks which match the single shot used for meta-training.

**Oracle**. The penalty for not having ground truth labels to construct near-optimal tasks ranges from substantial to severe, depending on the difficulty of the downstream task. Easier downstream tasks (which have fewer classes and/or more supervision) incur less of a penalty. We conjecture that with such tasks, the difference in the usefulness of the priors matters less since the downstream task-specific evidence has more power to shape the posterior.

Table 2: Results of unsupervised learning on miniImageNet and CelebA images, averaged over 1000 downstream human-designed tasks. CACTUs experiments use $k = 500$ for each of $P = 50$ partitions. Embedding cluster matching uses the same $k$. For complete results with confidence intervals, see Tables 9 and 10 in Appendix F.

| Algorithm | (way, shot) | miniImageNet (5, 1) | (5, 5) | (5, 20) | (5, 50) | CelebA (2, 5) |
|---|---|---|---|---|---|---|
| Training from scratch | | 27.59% | 38.48% | 51.53% | 59.63% | 63.19% |
| BiGAN $k_{nn}$-nearest neighbors | | 25.56% | 31.10% | 37.31% | 43.60% | 56.15% |
| BiGAN linear classifier | | 27.08% | 33.91% | 44.00% | 50.41% | 58.44% |
| BiGAN MLP with dropout | | 22.91% | 29.06% | 40.06% | 48.36% | 56.26% |
| BiGAN cluster matching | | 24.63% | 29.49% | 33.89% | 36.13% | 56.20% |
| BiGAN CACTUs-MAML (ours) | | 36.24% | 51.28% | 61.33% | 66.91% | **74.98%** |
| BiGAN CACTUs-ProtoNets (ours) | | 36.62% | 50.16% | 59.56% | 63.27% | 65.58% |
| DeepCluster $k_{nn}$-nearest neighbors | | 28.90% | 42.25% | 56.44% | 63.90% | 61.47% |
| DeepCluster linear classifier | | 29.44% | 39.79% | 56.19% | 65.28% | 59.57% |
| DeepCluster MLP with dropout | | 29.03% | 39.67% | 52.71% | 60.95% | 60.65% |
| DeepCluster cluster matching | | 22.20% | 23.50% | 24.97% | 26.87% | 51.51% |
| DeepCluster CACTUs-MAML (ours) | | **39.90%** | **53.97%** | **63.84%** | **69.64%** | 73.79% |
| DeepCluster CACTUs-ProtoNets (ours) | | **39.18%** | **53.36%** | 61.54% | 63.55% | **74.15%** |
| Oracle-MAML (control) | | 46.81% | 62.13% | 71.03% | 75.54% | 87.10% |
| Oracle-ProtoNets (control) | | 46.56% | 62.29% | 70.05% | 72.04% | 85.13% |

**Task construction ablation**. As seen in Tables 3 and 4, CACTUs-MAML consistently outperforms Hyperplanes-MAML with any margin. We hypothesize that this is due to the issues with zero-margin Hyperplanes-MAML pointed out in Appendix C, and the fact that nonzero-margin Hyperplanes-MAML is able to use less of the meta-training split to generate tasks than CACTUs-MAML is. We find that using multiple partitions for CACTUs-MAML, while beneficial, is not strictly necessary. Using non-zero margin with Hyperplanes-MAML is crucial for miniImageNet, but not for Omniglot. We conjecture that the enforced degree of separation between classes is needed for miniImageNet because of the dataset's high diversity. Meta-learning on random tasks or tasks derived from pixel-space clustering (Table 3) results in a prior that is much less useful than any other considered algorithm, including a random network initialization; evidently, practicing badly is worse than not practicing at all.

**Note on overfitting**. Because of the combinatorially many unsupervised tasks we can create from multiple partitions of the dataset, we do not observe substantial overfitting to the unsupervised meta-training tasks. However, we observe that meta-training performance is sometimes worse than meta-test time performance, which is likely due to a portion of the automatically generated tasks being based on nonsensical clusters (for examples, see Figure 2). Additionally, we find that, with a few exceptions, using multiple partitions has a regularizing effect on the meta-learner: a diverse task set reduces overfitting to the meta-training tasks and increases the applicability of the learned prior.

Table 3: Ablation study of task construction methods on Omniglot. For a more complete set of results with confidence intervals, see Table 8 in Appendix F.

| Algorithm | (way, shot) | (5, 1) | (5, 5) | (20, 1) | (20, 5) |
|---|---|---|---|---|---|
| Random-MAML, $P = 2400$, $k = 500$ | | 25.99% | 25.74% | 6.51% | 6.74% |
| Pixels CACTUs-MAML, $P = 1$, $k = 500$ | | 30.55% | 40.19% | 12.05% | 19.01% |
| ACAI Hyperplanes-MAML, $P = 2400$, $m = 0$ | | 62.34% | 81.81% | 39.30% | 63.18% |
| ACAI Hyperplanes-MAML, $P = 2400$, $m = 1.2$ | | 62.44% | 83.20% | 41.86% | 65.23% |
| ACAI CACTUs-MAML, $P = 1$, $k = 500$ | | 66.49% | 85.60% | 45.04% | 69.14% |
| ACAI CACTUs-MAML, $P = 100$, $k = 500$ | | **68.84%** | **87.78%** | **48.09%** | **73.36%** |
| BiGAN Hyperplanes-MAML, $P = 2400$, $m = 0$ | | 53.60% | 74.60% | 29.02% | 50.77% |
| BiGAN Hyperplanes-MAML, $P = 2400$, $m = 0.5$ | | 53.18% | 73.55% | 29.98% | 50.14% |
| BiGAN CACTUs-MAML, $P = 1$, $k = 500$ | | 55.92% | 76.28% | 32.44% | 54.22% |
| BiGAN CACTUs-MAML, $P = 100$, $k = 500$ | | 58.18% | 78.66% | 35.56% | 58.62% |

Table 4: Ablation study of task construction methods on miniImageNet. For a more complete set of results with confidence intervals, see Table 9 in Appendix F.

| Algorithm | (way, shot) | (5, 1) | (5, 5) | (5, 20) | (5, 50) |
|---|---|---|---|---|---|
| BiGAN Hyperplanes-MAML, $P = 4800, m = 0$ | | 20.00% | 20.00% | 20.00% | 20.00% |
| BiGAN Hyperplanes-MAML, $P = 4800, m = 0.9$ | | 29.67% | 41.92% | 51.32% | 54.72% |
| BiGAN CACTUs-MAML, $P = 1, k = 500$ | | 37.75% | 52.59% | 62.70% | 67.98% |
| BiGAN CACTUs-MAML, $P = 50, k = 500$ | | 36.24% | 51.28% | 61.33% | 66.91% |
| DeepCluster Hyperplanes-MAML, $P = 4800, m = 0$ | | 20.02% | 20.01% | 20.00% | 20.01% |
| DeepCluster Hyperplanes-MAML, $P = 4800, m = 0.1$ | | 35.85% | 49.54% | 60.68% | 65.55% |
| DeepCluster CACTUs-MAML, $P = 1, k = 500$ | | 38.75% | 52.73% | 62.72% | 67.77% |
| DeepCluster CACTUs-MAML, $P = 50, k = 500$ | | **39.90%** | **53.97%** | **63.84%** | **69.64%** |

## 5 DISCUSSION

We demonstrate that meta-learning on tasks produced using simple mechanisms based on embeddings improves upon the utility of these representations in learning downstream, human-specified tasks. We empirically show that this holds across benchmark datasets and tasks in the few-shot classification literature (Santoro et al., 2016; Ravi & Larochelle, 2017; Finn et al., 2018), task difficulties, and embedding learning methods while fixing key hyperparameters across all experiments.

In a sense, CACTUs can be seen as a facilitating interface between an embedding learning method and a meta-learning algorithm. As shown in the results, the meta-learner's performance significantly depends on the nature and quality of the task-generating embeddings. We can expect our method to yield better performance as the methods that produce these embedding functions improve, becoming better suited for generating diverse yet distinctive clusterings of the data. However, the gap between unsupervised and supervised meta-learning will likely persist because, with the latter, the meta-training task distribution is human-designed to mimic the expected evaluation task distribution as much as possible. Indeed, to some extent, supervised meta-learning algorithms offload the effort of designing and tuning algorithms onto the effort of designing and tuning task distributions. With its evaluation-agnostic task generation, CACTUs-based meta-learning trades off performance in specific use-cases for broad applicability and the ability to train on unlabeled data. In principle, CACTUs-based meta-learning may outperform supervised meta-learning when the latter is trained on a misaligned task distribution. We leave this investigation to future work.

While we have demonstrated that $k$-means is a broadly useful mechanism for constructing tasks from embeddings, it is unlikely that combinations of $k$-means clusters in learned embedding spaces are universal approximations of arbitrary class definitions. An important direction for future work is to find examples of datasets and human-designed tasks for which CACTUs-based meta-learning results in ineffective downstream learning. This will result in better understanding of the practical scope of applicability for our method, and spur further development in automatic task construction mechanisms for unsupervised meta-learning.

A potential concern of our experimental evaluation is that MNIST, Omniglot, and miniImageNet exhibit particular structure in the underlying class distribution (i.e., perfectly balanced classes), since they were designed to be supervised learning benchmarks. In more practical applications of machine learning, such structure would likely not exist. Our CelebA results indicate that CACTUs is effective even in the case of a dataset without neatly balanced classes or attributes. An interesting direction for future work is to better characterize the performance of CACTUs and other unsupervised pre-training methods with highly-unstructured, unlabeled datasets.

Since MAML and ProtoNets produce nothing more than a learned representation, our method can be viewed as deriving, from a previous unsupervised representation, a new representation particularly suited for learning downstream tasks. Beyond visual classification tasks, the notion of using unsupervised pre-training is generally applicable to a wide range of domains, including regression, speech (Oord et al., 2018), language (Howard & Ruder, 2018), and reinforcement learning (Shelhamer et al., 2017). Hence, our unsupervised meta-learning approach has the potential to improve unsupervised representations for a variety of such domains, an exciting avenue for future work.

ACKNOWLEDGMENTS

We thank Kelvin Xu, Richard Zhang, Brian Cheung, Ben Poole, Aäron van den Oord, Luke Metz, Siddharth Reddy, and the anonymous reviewers for feedback on an early draft of this paper.

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

## APPENDIX A  THE EMBEDDING LEARNING ZOO

We evaluate four distinct methods from prior work for learning the task-generating embeddings.

In adversarially constrained autoencoder interpolation (ACAI), a convolutional autoencoder's pixel-wise $L^2$ loss is regularized with a term encouraging meaningful interpolations in the latent space (Berthelot et al., 2018). Specifically, a critic network takes as input a synthetic image generated from a convex combination of the latents of two dataset samples, and regresses to the mixing factor. The decoder of the autoencoder and the generator for the critic are one and the same. The regularization term is minimized when the autoencoder fools the critic into predicting that the synthetic image is a real sample.

The bidirectional GAN (BiGAN) is an instance of a generative-adversarial framework in which the generator produces both synthetic image and embedding from real embedding and image, respectively (Donahue et al., 2017; Dumoulin et al., 2017). Discrimination is done in joint image-embedding space.

The DeepCluster method does discriminative clustering by alternating between clustering the features of a convolutional neural network and using the clusters as labels to optimize the network weights via backpropagating a standard classification loss (Caron et al., 2018).

The InfoGAN framework conceptually decomposes the generator's input into a latent code and incompressible noise (Chen et al., 2016). The structure of the latent code is hand-specified based on knowledge of the dataset. The canonical GAN minimax objective is regularized with a mutual information term between the code and the generated image. In practice, this term is optimized using variational inference, involving the approximation of the posterior with an auxiliary distribution $Q(\text{code}|\text{image})$ parameterized by a recognition network.

Whereas ACAI explicitly optimizes pixel-wise reconstruction error, BiGAN only encourages the fidelity of generated image and latent samples with respect to their respective prior distributions. While InfoGAN also encourages the fidelity of generated images, it leverages domain-specific knowledge to impose a favorable structure on the embedding space and information-theoretic methods for optimization. DeepCluster departs from the aforementioned methods in that it is not concerned with generation or decoding, and only seeks to learn general-purpose visual features by way of end-to-end discriminative clustering.

## APPENDIX B  DATASET INFORMATION

The Omniglot dataset consists of 1623 characters each with 20 hand-drawn examples. Ignoring the alphabets from which the characters originate, we use 1100, 100, and 423 characters for our meta-training, meta-validation, and meta-testing splits. The miniImageNet dataset consists of 100 classes each with 600 examples. The images are predominantly natural and realistic. We use the same meta-training/meta-validation/meta-testing splits of 64/16/20 classes as proposed by Ravi & Larochelle (2017). The CelebA dataset includes 202,599 facial images of celebrities and 40 binary attributes that annotate every image. We follow the prescribed 162,770/19,867/19,962 data split.

For Omniglot and miniImageNet, supervised meta-learning tasks and evaluation tasks are constructed exactly as detailed in Section 2.2: for an $N$-way $K$-shot task with $Q$ queries per class, we sample $N$ classes from the data split and $K + Q$ datapoints per class, labeling the task's data with a random permutation of $N$ one-hot vectors.

For CelebA, we consider binary classification tasks (i.e., 2-way), each defined by 3 attributes and an ordering of 3 Booleans, one for each attribute. Every image in a task-specific class shares all task-specific attributes with each other and none with images in the other class. For example, the task illustrated in Figure 2 involves distinguishing between images whose subjects satisfy `not Sideburns`, `Straight Hair`, and `not Young`, and those whose subjects satisfy `Sideburns`, `not Straight Hair`, and `Young`. To keep with the idea of having distinct classes for meta-training and meta-testing, we split the task-defining attributes. For the supervised meta-learning oracle, we construct meta-training tasks from the first 20 attributes (when alphabetically ordered), meta-validation tasks from the next 10, and meta-testing tasks from the last 10. Discarding tasks with too few examples in either class, this results in 4287, 391, and 402 task

prototypes (but many more possible tasks). We use the same meta-test time tasks to evaluate the unsupervised methods. We only consider assessment with 5-shot tasks because, given that there are multiple attributes other than the task-defining ones, any 1-shot task is likely to be ill-defined.

## APPENDIX C    TASK CONSTRUCTION VIA RANDOM HYPERPLANES

Given a set of embedding points $\{\mathbf{z}_i\}$ in a space $\mathcal{Z}$, a simple way of defining a partition $\mathcal{P} = \{\mathcal{C}_c\}$ on $\{\mathbf{z}_i\}$ is to use random hyperplanes to slice $\mathcal{Z}$ into subspaces and assign the embeddings that lie in the $c$-th subspace to subset $\mathcal{C}_c$. However, a hyperplane slicing can group together two arbitrarily far embeddings, or separate two arbitrarily close ones; given our assumption that good embedding spaces have a semantically meaningful metric, this creates ill-defined classes. This problem can be partially alleviated by extending the hyperplane boundaries with a non-zero margin, as empirically shown in Section 4.2.

We now describe how to generate tasks via random hyperplanes in the embedding space. We first describe a procedure to generate a partition $\mathcal{P}$ of the set of embeddings $\{\mathbf{z}_i\}$ for constructing meta-training tasks. A given hyperplane slices the embedding space into two, so for an $N$-way task, we need $H = \lceil \log_2 N \rceil$ hyperplanes to define sufficiently many subsets/classes for a task. To randomly define a hyperplane in $d$-dimensional embedding space, we sample a normal vector $\mathbf{n}$ and a point on the plane $\mathbf{z}_0$, each with $d$ elements. For an embedding point $\mathbf{z}$, the signed point-plane distance is given by $\frac{\mathbf{n}}{|\mathbf{n}|_2} \cdot (\mathbf{z} - \mathbf{z}_0)$. Defining $H$ hyperplanes in this manner, we discard embeddings for which the signed point-plane distance to any of the $H$ hyperplanes lies within $(-m, m)$, where $m$ is a desired margin. The $H$ hyperplanes collectively define $2^H$ subspaces. We assign embedding points in the $c$-th subspace to subset $\mathcal{C}_c$. We define the partition as $\mathcal{P} = \{\mathcal{C}_c\}$. We prune subsets that do not have at least $R = K_{\text{m-tr}} + Q$ members, and check that the partition has at least $N$ remaining subsets; if not, we reject the partition and restart the procedure. After obtaining partitions $\{\mathcal{P}_p\}$, meta-training tasks can be generated by following Algorithm 1 from Line 4.

In terms of practical implementation, we pre-compute 1000 hyperplanes and pruned pairs of subsets of $\{\mathbf{z}_i\}$. We generate partitions by sampling combinations of the hyperplanes and taking intersections of their associated subsets to define the elements of the partition. We determine the number of partitions needed for a given Hyperplanes-MAML run by the number of meta-training tasks desired for the meta-learner: we fix 100 tasks per partition.

## APPENDIX D    MNIST EXPERIMENTS

The MNIST dataset consists of 70,000 hand-drawn examples of the 10 numerical digits. Our split respects the original MNIST 60,000/10,000 training/testing split. We assess on 10-way classification tasks. This setup results in examples from all 10 digits being present for both meta-training and meta-testing, making the probem setting essentially equivalent to that of semi-supervised learning sans a fixed permutation of the labels. The MNIST scenario is thus a special case of the problem setting considered in the rest of the paper. For MNIST, we only experiment with MAML as the meta-learning algorithm.

For ACAI and InfoGAN we constructed the meta-validation split from the last 5,000 examples of the meta-training split; for BiGAN this figure was 10,000. After training the ACAI model and inferring embeddings, manually assigning labels to 10 clusters by inspection results in a classification accuracy of 96.00% on the testing split. As the ACAI authors observe, we found it important to whiten the ACAI embeddings before clustering. The same metric for the InfoGAN embedding (taking an argmax over the categorical dimensions instead of actually running clustering) is 96.83%. Note that these results are an upper-bound for embedding cluster matching. To see this, consider the 10-way 1-shot scenario. 1 example sampled from each cluster is insufficient to guarantee the optimal label for that cluster; 1 example sampled from each label is not guaranteed to each end up in the optimal category.

Aside from CACTUs-MAML, embedding $k_{\text{nn}}$-nearest neighbors, embedding linear classifier, and embedding direct clustering, we also ran CACTUs-MAML on embeddings instead of raw images, using a simple model with 2 hidden layers with 64 units each and ReLU activation, and all other MAML hyperparameters being the same as in Table 5.

Departing from the fixed $k = 500$ used for all other datasets, we deliberately use $k = 10$ to better understand the limitations of CACTUs-MAML. The results can be seen in Table 7 in Appendix B. In brief, with the better embeddings (ACAI and InfoGAN), there is only little benefit of CACTUs-MAML over embedding cluster matching. Additionally, even in the best cases, CACTUs-MAML falls short of state-of-the-art semi-supervised learning methods.

## APPENDIX E  HYPERPARAMETERS AND ARCHITECTURES

### E.1  MAML

Table 5: MAML hyperparameter summary.

| Hyperparameter | MNIST | Omniglot | miniImageNet | CelebA |
|---|---|---|---|---|
| Input size | $28 \times 28$ | $28 \times 28$ | $84 \times 84 \times 3$ | $84 \times 84 \times 3$ |
| Outer (meta) learning rate | 0.001 | 0.001 | 0.001 | 0.001 |
| Inner learning rate | 0.05 | 0.05 | 0.05 | 0.05 |
| Task batch size | 8 | 8 | 8 | 8 |
| Inner adaptation steps (meta-training) | 5 | 5 | 5 | 5 |
| Meta-training iterations | 30,000 | 30,000 | 60,000 | 60,000 |
| Adaptation steps (evaluation) | 50 | 50 | 50 | 50 |
| Classes per task (meta-training) | 10 | 20 | 5 | 2 |
| Shots per class (meta-training) | 1 | 1 | 1 | 1 |
| Queries per class | 5 | 5 | 5 | 5 |

For MNIST and Omniglot we use the same 4-block convolutional architecture as used by Finn et al. (2017) for their Omniglot experiments, but with 32 filters (instead of 64) for each convolutional layer for consistency with the model used for miniImageNet and CelebA, which is the same as what Finn et al. (2017) used for their miniImageNet experiments. When evaluating the meta-learned 20-way Omniglot model with 5-way tasks, we prune the unused output dimensions. The outer optimizer is Adam (Kingma & Ba, 2014), and the inner optimizer is SGD. We build on the authors' publicly available codebase found at `https://github.com/cbfinn/maml`.

When using batch normalization (Ioffe & Szegedy, 2015) to process a task's training or query inputs, we observe that using only 1 query datapoint per class can allow the model to exploit batch statistics, learning a strategy analogous to a process of elimination that causes significant, but spurious, improvement in accuracy. To mitigate this, we fix 5 queries per class for every task's evaluation phase, meta-training or meta-testing.

### E.2  PROTONETS

Table 6: ProtoNets hyperparameter summary.

| Hyperparameter | Omniglot | miniImageNet | CelebA |
|---|---|---|---|
| Input size | $28 \times 28$ | $84 \times 84 \times 3$ | $84 \times 84 \times 3$ |
| Learning rate | 0.001 | 0.001 | 0.001 |
| Task batch size | 1 | 1 | 1 |
| Training iterations | 30,000 | 60,000 | 60,000 |
| Classes per task (meta-training) | 20 | 5 | 2 |
| Shots per class (meta-training) | 1 | 1 | 1 |
| Queries per class (meta-training/meta-testing) | 15/5 | 15/5 | 15/5 |

For the three considered datasets we use the same architecture as used by Snell et al. (2017) for their Omniglot and miniImageNet experiments. This is a 4-block convolutional architecture with each

block consisting of a convolutional layer with 64 $3 \times 3$ filters, stride 1, and padding 1, followed by BatchNorm, ReLU activation, and $2 \times 2$ MaxPooling. The ProtoNets embedding is simply the flattened output of the last block. We follow the authors and use the Adam optimizer, but do not use a learning rate scheduler. We build upon the authors' publicly available codebase found at `https://github.com/jakesnell/prototypical-networks`.

### E.3 CACTUs

For Omniglot, miniImageNet, and CelebA we fix the number of clusters $k$ to be 500. For Omniglot we choose the number of partitions $P = 100$, but in the interest of keeping runtime manageable, choose $P = 50$ for miniImageNet and CelebA.

### E.4 USE OF UNSUPERVISED LEARNING METHODS

ACAI (Berthelot et al., 2018): We run ACAI for MNIST and Omniglot. We pad the images by 2 and use the authors' architecture. We use a 256-dimensional embedding for all datasets. We build upon the authors' publicly available codebase found at `https://github.com/brain-research/acai`.

We unsuccessfully try running ACAI on $64 \times 64$ miniImageNet and CelebA. To facilitate this input size, we add one block consisting of two convolutional layers (512 filters each) and one down-sampling/upsampling layer to the encoder and decoder. However, because of ACAI's pixel-wise reconstruction loss, for these datasets the ACAI embedding prioritizes information about the few "features" that dominate the reconstruction pixel count, resulting in clusters that only corresponded to a limited range of factors, such as background color and pose. For curiosity's sake, we tried running meta-learning on tasks derived from these uninteresting clusters anyways, and found that the meta-learner quickly produced a learning procedure that obtained high accuracy on the meta-training tasks. However, this learned prior was not useful for solving downstream tasks.

BiGAN (Donahue et al., 2017): For MNIST, we follow the BiGAN authors and specify a uniform 50-dimensional prior on the unit hypercube for the latent. The BiGAN authors use a 200-dimensional version of the same prior for their ImageNet experiments, so we follow suit for Omniglot, miniImageNet, and CelebA. For MNIST and Omniglot, we use the permutation-invariant architecture (i.e. fully connected layers only) used by the authors for their MNIST results; for miniImageNet and CelebA, we randomly crop to $64 \times 64$ and use the AlexNet-inspired architecture used by Donahue et al. (2017) for their ImageNet results. We build upon the authors' publicly available codebase found at `https://github.com/jeffdonahue/bigan`.

DeepCluster (Caron et al., 2018): We run DeepCluster for miniImageNet and CelebA, which we respectively randomly crop and resize to $64 \times 64$. We modify the first layer of the AlexNet architecture used by the authors to accommodate this input size. We follow the authors and use the input to the (linear) output layer as the embedding. These are 4096-dimensional, so we follow the authors and apply PCA to reduce the dimensionality to 256, followed by whitening. We build upon the authors' publicly available codebase found at `https://github.com/facebookresearch/deepcluster`.

InfoGAN (Chen et al., 2016): We only run InfoGAN for MNIST. We follow the InfoGAN authors and specify the product of a 10-way categorical distribution and a 2-dimensional uniform distribution as the latent code. We use the authors' architecture. Given an image, we use the recognition network to obtain its embedding. We build upon the authors' publicly available codebase found at `https://github.com/openai/InfoGAN`.

## APPENDIX F  EXPERIMENTAL RESULTS

This section containsfull experimental results for the MNIST, Omniglot, miniImageNet, and CelebA datasets, including consolidated versions of the tables found in the main text. The metric is classification accuracy averaged over 1000 tasks based on human-specified labels of the testing split, with 95% confidence intervals. $d$: dimensionality of embedding, $h$: number of hidden units in a fully connected layer, $k$: number of clusters in a partition, $P$: number of partitions used during meta-learning, $m$: margin on boundary-defining hyperplanes.

Table 7: MNIST digit classification results averaged over 1000 tasks. $\pm$ denotes a 95% confidence interval. $k$: number of clusters in a partition, $P$: number of partitions used during meta-learning

| Algorithm | (way, shot) | (10, 1) | (10, 5) | (10, 10) |
|---|---|---|---|---|
| *ACAI, $d = 256$* | | | | |
| Embedding $k_{nn}$-nearest neighbors | | 74.49 ± 0.82 % | 88.80 ± 0.27 % | 91.90 ± 0.17 % |
| Embedding linear classifier | | 76.53 ± 0.81 % | 92.17 ± 0.25 % | 94.58 ± 0.15 % |
| Embedding cluster matching, $k = 10$ | | 91.28 ± 0.58 % | 95.92 ± 0.16 % | 96.01 ± 0.12 % |
| CACTUs-MAML on images (ours), $P = 1, k = 10$ | | 92.66 ± 0.34 % | 96.08 ± 0.12 % | 96.29 ± 0.12 % |
| CACTUs-MAML on embeddings (ours), $P = 1, k = 10$ | | 94.77 ± 0.28 % | 96.56 ± 0.11 % | 96.80 ± 0.11 % |
| *BiGAN, $d = 50$* | | | | |
| Embedding $k_{nn}$-nearest neighbors | | 29.25 ± 0.83 % | 44.59 ± 0.44 % | 51.98 ± 0.30 % |
| Embedding linear classifier | | 30.86 ± 0.89 % | 51.69 ± 0.44 % | 60.70 ± 0.31 % |
| Embedding cluster matching, $k = 10$ | | 33.72 ± 0.54 % | 50.21 ± 0.36 % | 52.95 ± 0.34% |
| CACTUs-MAML on images (ours), $P = 1, k = 10$ | | 43.75 ± 0.46 % | 62.20 ± 0.33 % | 68.38 ± 0.29 % |
| CACTUs-MAML on images (ours), $P = 100, k = 10$ | | 49.73 ± 0.45 % | 77.05 ± 0.30 % | 83.90 ± 0.24 % |
| CACTUs-MAML on embeddings (ours), $P = 1, k = 10$ | | 36.33 ± 0.48 % | 51.78 ± 0.34 % | 57.41 ± 0.30 % |
| CACTUs-MAML on embeddings (ours), $P = 100, k = 10$ | | 37.32 ± 0.41 % | 60.74 ± 0.34 % | 67.34 ± 0.30 % |
| *InfoGAN, $d = 12$* | | | | |
| Embedding $k_{nn}$-nearest neighbors | | 94.53 ± 0.51 % | 96.05 ± 0.17 % | 96.24 ± 0.12 % |
| Embedding linear classifier | | 95.78 ± 0.42 % | 96.61 ± 0.21 % | 96.85 ± 0.11 % |
| Embedding cluster matching, $k = 10$ | | 93.42 ± 0.57 % | 96.97 ± 0.15 % | 96.99 ± 0.10 % |
| CACTUs-MAML on images (ours), $P = 1, k = 10$ | | 95.30 ± 0.23 % | **97.18 ± 0.10 %** | **97.28 ± 0.10 %** |
| CACTUs-MAML on images (ours), $P = 100, k = 10$ | | 96.08 ± 0.19 % | **97.22 ± 0.10 %** | **97.31 ± 0.09 %** |
| CACTUs-MAML on embeddings (ours), $P = 1, k = 10$ | | **96.69 ± 0.17 %** | **97.13 ± 0.10 %** | **97.23 ± 0.10 %** |
| CACTUs-MAML on embeddings (ours), $P = 100, k = 10$ | | 96.48 ± 0.17 % | 97.08 ± 0.10 % | **97.22 ± 0.10 %** |
| *Supervised pre-training* | | | | |
| Oracle-MAML (control) | | 97.31 ± 0.17 % | 98.51 ± 0.07 % | 98.51 ± 0.07 % |

Table 8: Omniglot character classification results averaged over 1000 tasks. $\pm$ denotes a 95% confidence interval. $d$: dimensionality of embedding, $h$: number of hidden units in a fully connected layer, $k$: number of clusters in a partition, $P$: number of partitions used during meta-learning, $m$: margin on boundary-defining hyperplanes.

| Algorithm | (way, shot) | (5, 1) | (5, 5) | (20, 1) | (20, 5) |
|---|---|---|---|---|---|
| *Baselines* | | | | | |
| Training from scratch | | 52.50 ± 0.84 % | 74.78 ± 0.69 % | 24.91 ± 0.33 % | 47.62 ± 0.44 % |
| Random-MAML, $P = 2400$, $k = 500$ | | 25.99 ± 0.73 % | 25.74 ± 0.69 % | 6.51 ± 0.18 % | 6.74 ± 0.18 % |
| Pixels CACTUs-MAML, $P = 1$, $k = 500$ | | 30.55 ± 0.63 % | 40.19 ± 0.71 % | 12.05 ± 0.23 % | 19.01 ± 0.29 % |
| *ACAI, $d = 256$* | | | | | |
| Embedding $k_{nn}$-nearest neighbors | | 57.46 ± 1.35 % | 81.16 ± 0.57 % | 39.73 ± 0.38 % | 66.38 ± 0.36 % |
| Embedding linear classifier | | 61.08 ± 1.32 % | 81.82 ± 0.58 % | 43.20 ± 0.69 % | 66.33 ± 0.36 % |
| Embedding MLP with dropout, $h = 128$ | | 51.95 ± 0.82 % | 77.20 ± 0.65 % | 30.65 ± 0.39 % | 58.62 ± 0.41 % |
| Embedding cluster matching, $k = 500$ | | 54.94 ± 0.85 % | 71.09 ± 0.77 % | 32.19 ± 0.40 % | 45.93 ± 0.40 % |
| Hyperplanes-MAML (ours), $P = 2400$, $m = 0$ | | 62.34 ± 0.82 % | 81.81 ± 0.60 % | 39.30 ± 0.37 % | 63.18 ± 0.38 % |
| Hyperplanes-MAML (ours), $P = 2400$, $m = 1.2$ | | 62.44 ± 0.82 % | 83.20 ± 0.58 % | 41.86 ± 0.38 % | 65.23 ± 0.37 % |
| CACTUs-MAML (ours), $P = 1$, $k = 500$ | | 66.49 ± 0.80 % | 85.60 ± 0.53 % | 45.04 ± 0.41 % | 69.14 ± 0.36 % |
| CACTUs-MAML (ours), $P = 100$, $k = 500$ | | **68.84 ± 0.80 %** | **87.78 ± 0.50 %** | **48.09 ± 0.41 %** | **73.36 ± 0.34 %** |
| CACTUs-ProtoNets (ours), $P = 100$, $k = 500$ | | **68.12 ± 0.84 %** | 83.58 ± 0.61 % | **47.75 ± 0.43 %** | 66.27 ± 0.37 % |
| *BiGAN, $d = 200$* | | | | | |
| Embedding $k_{nn}$-nearest neighbors | | 49.55 ± 1.27 % | 68.06 ± 0.71 % | 27.37 ± 0.33 % | 46.70 ± 0.36 % |
| Embedding linear classifier | | 48.28 ± 1.25 % | 68.72 ± 0.66 % | 27.80 ± 0.61 % | 45.82 ± 0.37 % |
| Embedding MLP with dropout, $h = 128$ | | 40.54 ± 0.79 % | 62.56 ± 0.79 % | 19.92 ± 0.32 % | 40.71 ± 0.40 % |
| Embedding cluster matching, $k = 500$ | | 43.96 ± 0.80 % | 58.62 ± 0.78 % | 21.54 ± 0.32 % | 31.06 ± 0.37 % |
| Hyperplanes-MAML (ours), $P = 2400$, $m = 0$ | | 53.60 ± 0.82 % | 74.60 ± 0.69 % | 29.02 ± 0.33 % | 50.77 ± 0.39 % |
| Hyperplanes-MAML (ours), $P = 2400$, $m = 0.5$ | | 53.18 ± 0.81 % | 73.55 ± 0.69 % | 29.98 ± 0.35 % | 50.14 ± 0.38 % |
| CACTUs-MAML (ours), $P = 1$, $k = 500$ | | 55.92 ± 0.80 % | 76.28 ± 0.65 % | 32.44 ± 0.35 % | 54.22 ± 0.39 % |
| CACTUs-MAML (ours), $P = 100$, $k = 500$ | | 58.18 ± 0.81 % | 78.66 ± 0.65 % | 35.56 ± 0.36 % | 58.62 ± 0.38 % |
| CACTUs-ProtoNets (ours), $P = 100$, $k = 500$ | | 54.74 ± 0.82 % | 71.69 ± 0.73 % | 33.40 ± 0.37 % | 50.62 ± 0.39 % |
| *Supervised meta-learning* | | | | | |
| Oracle-MAML (control) | | 94.46 ± 0.35 % | 98.83 ± 0.12 % | 84.60 ± 0.32 % | 96.29 ± 0.13 % |
| Oracle-ProtoNets (control) | | 98.35 ± 0.22 % | 99.58 ± 0.09 % | 95.31 ± 0.18 % | 98.81 ± 0.07 % |

† Result used 64 filters per convolutional layer, 3× data augmentation, and folded the validation set into the training set after hyperparameter tuning.

Table 9: miniImageNet object classification results averaged over 1000 tasks. ± denotes a 95% confidence interval. $d$: dimensionality of embedding, $h$: number of hidden units in a fully connected layer, $k$: number of clusters in a partition, $P$: number of partitions used during meta-learning, $m$: margin on boundary-defining hyperplanes.

| Algorithm | (way, shot) | (5, 1) | (5, 5) | (5, 20) | (5, 50) |
|---|---|---|---|---|---|
| *Baselines* | | | | | |
| Training from scratch | | 27.59 ± 0.59 % | 38.48 ± 0.66 % | 51.53 ± 0.72 % | 59.63 ± 0.74 % |
| *BiGAN, $d = 200$* | | | | | |
| Embedding $k_m$-nearest neighbors | | 25.56 ± 1.08 % | 31.10 ± 0.63 % | 37.31 ± 0.40 % | 43.60 ± 0.37 % |
| Embedding linear classifier | | 27.08 ± 1.24 % | 33.91 ± 0.64 % | 44.00 ± 0.45 % | 50.41 ± 0.37 % |
| Embedding MLP with dropout, $h = 128$ | | 22.91 ± 0.54 % | 29.06 ± 0.63 % | 40.06 ± 0.72 % | 48.36 ± 0.71 % |
| Embedding cluster matching, $k = 500$ | | 24.63 ± 0.56 % | 29.49 ± 0.58 % | 33.89 ± 0.63 % | 36.13 ± 0.64 % |
| Hyperplanes-MAML (ours), $P = 4800$, $m = 0$ | | 20.00 ± 0.00 % | 20.00 ± 0.00 % | 20.00 ± 0.00 % | 20.00 ± 0.00 % |
| Hyperplanes-MAML (ours), $P = 4800$, $m = 0.9$ | | 29.67 ± 0.64 % | 41.92 ± 0.69 % | 51.32 ± 0.71 % | 54.72 ± 0.71 % |
| CACTUs-MAML (ours), $P = 1$, $k = 500$ | | 37.75 ± 0.74 % | 52.59 ± 0.75 % | 62.70 ± 0.68 % | 67.98 ± 0.68 % |
| CACTUs-MAML (ours), $P = 50$, $k = 500$ | | 36.24 ± 0.74 % | 51.28 ± 0.68 % | 61.33 ± 0.67 % | 66.91 ± 0.68 % |
| CACTUs-ProtoNets (ours), $P = 50$, $k = 500$ | | 36.62 ± 0.70 % | 50.16 ± 0.73 % | 59.56 ± 0.68 % | 63.27 ± 0.67 % |
| *DeepCluster, $d = 256$* | | | | | |
| Embedding $k_m$-nearest neighbors | | 28.90 ± 1.25 % | 42.25 ± 0.67 % | 56.44 ± 0.43 % | 63.90 ± 0.38 % |
| Embedding linear classifier | | 29.44 ± 1.22 % | 39.79 ± 0.64 % | 56.19 ± 0.43 % | 65.28 ± 0.34 % |
| Embedding MLP with dropout, $h = 128$ | | 29.03 ± 0.61 % | 39.67 ± 0.69 % | 52.71 ± 0.62 % | 60.95 ± 0.63 % |
| Embedding cluster matching, $k = 500$ | | 22.20 ± 0.50 % | 23.50 ± 0.52 % | 24.97 ± 0.54 % | 26.87 ± 0.55 % |
| Hyperplanes-MAML (ours), $P = 4800$, $m = 0$ | | 20.02 ± 0.06 % | 20.01 ± 0.01 % | 20.00 ± 0.01 % | 20.01 ± 0.02 % |
| Hyperplanes-MAML (ours), $P = 4800$, $m = 0.1$ | | 35.85 ± 0.66 % | 49.54 ± 0.72 % | 60.68 ± 0.69 % | 65.55 ± 0.66 % |
| CACTUs-MAML (ours), $P = 1$, $k = 500$ | | 38.75 ± 0.70 % | 52.73 ± 0.72 % | 62.72 ± 0.69 % | 67.77 ± 0.62 % |
| CACTUs-MAML (ours), $P = 50$, $k = 500$ | | **39.90 ± 0.74 %** | **53.97 ± 0.70 %** | **63.84 ± 0.70 %** | **69.64 ± 0.63 %** |
| CACTUs-ProtoNets (ours), $P = 50$, $k = 500$ | | **39.18 ± 0.71 %** | **53.36 ± 0.70 %** | 61.54 ± 0.68 % | 63.55 ± 0.64 % |
| *Supervised meta-learning* | | | | | |
| Oracle-MAML (control) | | 46.81 ± 0.77 % | 62.13 ± 0.72 % | 71.03 ± 0.69 % | 75.54 ± 0.62 % |
| Oracle-ProtoNets (control) | | 46.56 ± 0.76 % | 62.29 ± 0.71 % | 70.05 ± 0.65 % | 72.04 ± 0.60 % |

Table 10: CelebA facial attribute classification results averaged over 1000 tasks. $\pm$ denotes a 95% confidence interval. $d$: dimensionality of embedding, $h$: number of hidden units in a fully connected layer, $k$: number of clusters in a partition, $P$: number of partitions used during meta-learning.

| Algorithm (way, shot) | (2, 5) |
|---|---|
| *Baselines* | |
| Training from scratch | $63.19 \pm 1.06$ % |
| *BiGAN*, $d = 200$ | |
| Embedding $k_{nn}$-nearest neighbors | $56.15 \pm 0.89$ % |
| Embedding linear classifier | $58.44 \pm 0.90$ % |
| Embedding MLP with dropout, $h = 128$ | $56.26 \pm 0.94$ % |
| Embedding cluster matching, $k = 500$ | $56.20 \pm 1.00$ % |
| CACTUs-MAML (ours), $P = 50, k = 500$ | $\mathbf{74.98 \pm 1.02}$ % |
| CACTUs-ProtoNets (ours), $P = 50, k = 500$ | $65.58 \pm 1.04$ % |
| *DeepCluster*, $d = 256$ | |
| Embedding $k_{nn}$-nearest neighbors | $61.47 \pm 0.99$ % |
| Embedding linear classifier | $59.57 \pm 0.98$ % |
| Embedding MLP with dropout, $h = 128$ | $60.65 \pm 0.98$ % |
| Embedding cluster matching, $k = 500$ | $51.51 \pm 0.89$ % |
| CACTUs-MAML (ours), $P = 50, k = 500$ | $\mathbf{73.79 \pm 1.01}$ % |
| CACTUs-ProtoNets (ours), $P = 50, k = 500$ | $\mathbf{74.15 \pm 1.02}$ % |
| *Supervised meta-learning* | |
| Oracle-MAML (control) | $87.10 \pm 0.85$ % |
| Oracle-ProtoNets (control) | $85.13 \pm 0.92$ % |

## APPENDIX G    IMAGENET EXPERIMENTS

We investigate unsupervised meta-learning in the context of a larger unsupervised meta-training dataset by using the ILSVRC 2012 dataset's training split (Russakovsky et al., 2015), which is a superset of the miniImageNet dataset (including meta-validation and meta-testing data) consisting of 1000 classes and over 1,200,000 images. To facilitate comparison to the previous miniImageNet experiments, for meta-validation and meta-test we use the miniImageNet meta-validation and meta-test splits. To avoid task leakage, we hold out all data from these 36 underlying classes from the rest of the data to construct the meta-training split.

For CACTUs, we use the best-performing unsupervised learning method from the previous experiments, DeepCluster, to obtain the embeddings. Following Caron et al. (2018), we run DeepCluster using the VGG-16 architecture with a 256-dimensional feature space and 10,000 clusters on the meta-training data until the normalized mutual information between the data-cluster mappings of two consecutive epochs converges. To our knowledge, no prior works have yet been published on using MAML for ImageNet-sized meta-learning. We extend the standard convolutional neural network model class with residual connections (He et al., 2016), validate hyperparameters with supervised meta-learning, then use it for unsupervised meta-learning without further tuning. See Table 11 for MAML hyperparameters. The training from scratch, embedding $k_{nn}$-nearest neighbors, and embedding linear classifier algorithms are the same as they were in the previous sets of experiments. For Oracle-MAML, we generated tasks using the ground-truth 964 ImageNet meta-training classes. We also run semi-supervised MAML, with the meta-training tasks consisting of CACTUs-based tasks as well as tasks constructed from the 64 miniImageNet meta-training classes. The unsupervised/supervised task proportion split was fixed according to the ratio of the number of data available to each task proposal method. As before, the meta-learning methods only meta-learned on 1-shot tasks.

Table 11: MAML hyperparameter summary for ImageNet.

| Hyperparameter | Value |
|---|---|
| Input size | $224 \times 224$ |
| Outer (meta) learning rate | 0.0001 |
| Inner learning rate | 0.001 |
| Task batch size | 3 |
| Inner adaptation steps (meta-training) | 5 |
| Meta-training iterations | 240,000 |
| Adaptation steps (evaluation) | 100 |
| Classes per task (meta-training) | 5 |
| Shots per class (meta-training) | 1 |
| Queries per class | 5 |
| Residual blocks | 5 |
| Layers per residual block | 2 |

We find that the vastly increased amount of unlabeled meta-training data (in comparison to miniImageNet) results in significant increases for all methods over their counterparts in Table 9 (other than training from scratch, which does not use this data). We find that CACTUs-MAML slightly outperforms embedding linear classifier for the 1-shot test tasks, but that the linear classifier on top of the unsupervised embedding becomes better as the amount of test time supervision increases. Augmenting the unsupervised tasks with (a small number of) supervised tasks during meta-training results in slight improvement for the 1-shot test tasks. The lackluster performance of CACTUs-MAML is unsurprising insofar as meta-learning with large task spaces is still an open problem: higher shot Oracle-MAML only marginally stays ahead of the embedding linear classifier, which is not the case in the other, smaller-scale experiments. We expect that using a larger architecture in conjunction with MAML (such as Kim et al. (2018)) would result in increased performance for all methods based on MAML. Further, given the extensive degree to which unsupervised learning methods have been studied, we suspect that unsupervised task construction coupled with better meta-learning algorithms and architectures will result in improved performance on the entire unsupervised learning problem. We leave such investigation to future work.

Table 12: miniImageNet object classification results averaged over 1000 tasks, with ImageNet-scale meta-training. $\pm$ denotes a 95% confidence interval. $d$: dimensionality of embedding, $k$: number of clusters in a partition, $P$: number of partitions used during meta-learning.

| Algorithm | (way, shot) | (5, 1) | (5, 5) | (5, 20) | (5, 50) |
|---|---|---|---|---|---|
| *Baselines* | | | | | |
| Training from scratch | | $28.64 \pm 0.55$ % | $39.71 \pm 0.63$ % | $53.67 \pm 0.75$ % | $59.68 \pm 0.77$ % |
| *DeepCluster, $d = 256$* | | | | | |
| Embedding $k_{nn}$-nearest neighbors | | $50.17 \pm 0.61$ % | $69.34 \pm 0.51$ % | $79.81 \pm 0.43$ % | $84.72 \pm 0.34$ % |
| Embedding linear classifier | | $58.73 \pm 0.62$ % | $\mathbf{79.05} \pm \mathbf{0.44}$ % | $\mathbf{87.41} \pm \mathbf{0.31}$ % | $\mathbf{90.10} \pm \mathbf{0.27}$ % |
| CACTUs-MAML (ours), $P = 1, k = 10000$ | | $\mathbf{60.11} \pm \mathbf{0.74}$ % | $76.42 \pm 0.72$ % | $82.85 \pm 0.66$ % | $85.25 \pm 0.67$ % |
| *Semi-supervised meta-learning* | | | | | |
| Semi-supervised MAML | | $61.75 \pm 0.75$ % | $76.43 \pm 0.70$ % | $82.83 \pm 0.69$ % | $85.27 \pm 0.61$ % |
| *Supervised meta-learning* | | | | | |
| Oracle-MAML (control) | | $74.52 \pm 0.77$ % | $86.23 \pm 0.71$ % | $91.14 \pm 0.69$ % | $92.06 \pm 0.65$ % |