# OpenReview forum: "Unsupervised Learning via Meta-Learning"
_ICLR.cc/2019/Conference_

### Official Review · AnonReviewer2 · 2018-10-31
**Interesting approach but motivation is not clear.**

**Rating:** 6
**Confidence:** 3

**Review:**

This paper proposes to construct multiple classification tasks from unsupervised data.

Quality:
The detail of the proposed method is not mathematically presented and its performance is not theoretically analyzed.
Although the proposed method is empirically shown to be superior to other approaches, the motivation is not clearly presented.
Hence the overall quality of this paper is not high.

Clarity:
The readability of this paper is not high as it is redundant or unclear at several points.
For example, Sections 2.1, 2.3 and Sections 2.2, 2.4 can be integrated, respectively, and more mathematical details can be included instead.

Originality:
The proposal of constructing meta-learning based on unsupervised learning seems to be original.

Significance:
- The motivation is not clear. The proposed method artificially generates a number of classification tasks. But how to use such classifiers for artificially generated labels in real-world applications is not motivated.
  It is better to give a representative application, to which the proposed method fits.
- There is no theoretical analysis on the proposed method.
  For example, why is the first embedding step required? Clustering can be directly performed on the give dataset D = {x_i}.
- Although the paper discusses using unsupervised learning for meta-learning, only k-means is considered in the proposed method.
  There are a number of types of unsupervised learning, including other clustering algorithms and other tasks such as outlier detection, hence analyzing them is also interesting.
- The proposed method includes several hyper-parameters. But how to set them in practice it not clear.

Pros:
- An interesting approach to meta-learning is presented.

Cons:
- Motivation is not clear.
- There is no theoretical analysis.

---

> ### Author Response · Authors · 2018-11-08
> **[2/2] Additional responses to the reviewer’s points.**
>
> “Although the paper discusses using unsupervised learning for meta-learning, only k-means is considered in the proposed method.”
> We do consider multiple types of unsupervised learning: this is described in “Different embedding spaces” and “Task construction” in Section 4. For the embeddings, we consider and evaluate four unsupervised learning methods/objectives covering discriminative clustering, generative modeling, interpolation, and information maximization. For constructing tasks from embeddings, we consider and evaluate random sampling and hyperplane slicing in addition to k-means.
>
> “why is the first embedding step required? Clustering can be directly performed on the give dataset D = {x_i}.”
> Given that the x_i in our experiments are images, we believe it is clear from intuition that clustering on x_i would not work well: distance metrics in pixel-space do not correspond well to semantic meaning. We will add this as a comparison in the paper.
>
> “The proposed method includes several hyper-parameters. But how to set them in practice it not clear.”
> The hyperparameters associated with the embedding learning stage can be tuned on the unlabeled meta-validation split. For clustering, we fix the number of clusters k across all dataset/embedding/task difficulty combinations presented in the main text. We demonstrate that the number of partitions P is unimportant for our method: P=1 and P=50/100 (for miniImagenet and CelebA / Omniglot) both perform well (Section 4.2). There is ample justification of the hyperparameters used for the task construction: choose N, the number of classes in each task, by upper-bounding the number of classes expected to be seen in a downstream task, and choose K to be 1 (Section 4.1). As motivated in the first paragraph of Section 4.1, all other hyperparameters were selected based on prior work.
>
> >> “performance is not theoretically analyzed.”
> We found that, given the generality of the problem statement, it was difficult to make headway on theoretical analysis. We therefore opted to prioritize a solid experimental evaluation of the proposed method. Historically, theoretical analysis has not been a requirement for high-quality contributions in this community. There are numerous examples of high-quality, impactful papers devoid of theoretical analysis or guarantees presented at ICLR in recent years [1, 2, 3, 4, 5].
>
> References
> [1] Zoph et al. ICLR 2017, https://openreview.net/forum?id=r1Ue8Hcxg
> [2] Karras et al. ICLR 2018, https://openreview.net/forum?id=Hk99zCeAb
> [3] Jaderberg et al. ICLR 2017, https://openreview.net/forum?id=SJ6yPD5xg
> [4] Lazaridou et al. ICLR 2017, https://openreview.net/forum?id=Hk8N3Sclg
> [5] Ravi & Larochelle ICLR 2017, https://openreview.net/pdf?id=rJY0-Kcll

---

> ### Author Response · Authors · 2018-11-08
> **[1/2] Thank you for your review. Can you elaborate on your feedback?**
>
> Thank you for your time in reviewing our work! We would like to improve the paper based on your feedback, and would benefit from a few clarifications on your part.
>
> “The motivation is not clear. The proposed method artificially generates a number of classification tasks. But how to use such classifiers for artificially generated labels in real-world applications is not motivated. It is better to give a representative application, to which the proposed method fits.”
> Our evaluation and results are on real-world image classification tasks first proposed by prior work (Omniglot: [1], miniImageNet: [2], CelebA: [3]) and used by virtually all few-shot learning works in the last few years [1, 2, 3, 4, 5, 6, 7, 8, 9]. The test tasks are not artificially generated, but are real few-shot image classification tasks. We give the general use-case of our method in the last sentence of Section 2.1. We have clarified these points in our revision. Do you still find a lack of representative application? If so, do you have suggestions for better evaluation tasks?
>
> “The detail of the proposed method is not mathematically presented”, “ … more mathematical details can be included instead.”
> We have added the optimization objective of k-means to the paper. Are there any other parts of the method that require more formalism?
>
> “The readability of this paper is not high as it is redundant or unclear at several points.
> For example, Sections 2.1, 2.3 and Sections 2.2, 2.4 can be integrated, respectively, and more mathematical details can be included instead.”
> We have implemented your suggested re-organization to reduce redundancy. Which portions of the text, specifically, remain redundant or unclear?
>
> References
> [1] Santoro et al. ICML 2016, http://proceedings.mlr.press/v48/santoro16.pdf
> [2] Ravi & Larochelle ICLR 2017, https://openreview.net/pdf?id=rJY0-Kcll
> [3] Finn et al. NIPS 2018, https://arxiv.org/abs/1806.02817
> [4] Vinyals et al. NIPS 2016, https://papers.nips.cc/paper/6385-matching-networks-for-one-shot-learning
> [5] Munkhdalai et al. ICML 2017, https://arxiv.org/abs/1703.00837
> [6] Finn et al. ICML 2017, https://arxiv.org/abs/1703.03400
> [7] Snell et al NIPS 2017, https://papers.nips.cc/paper/6996-prototypical-networks-for-few-shot-learning
> [8] Oreshkin et al. NIPS 2018,
> https://arxiv.org/abs/1805.10123
> [9] Yoon et al. NIPS 2018, https://arxiv.org/abs/1806.03836

---

### Official Review · AnonReviewer3 · 2018-11-02
**Great paper tackling important problem with nice experiments**

**Rating:** 8
**Confidence:** 4

**Review:**

In this paper, the task of performing meta-learning based on the unsupervised dataset is considered. The high-level idea is to generate 'pseudo-labels' via clustering of the given dataset using existing unsupervised learning techniques. Then the meta-learning algorithm is trained to easily discriminate between such labels. This paper seems to be tackling an important problem that has not been addressed yet to my knowledge. While the proposed method/contribution is quite simple, it possesses great potential for future applications and deeper exploration. The empirical results look strong and tried to address important aspects of the algorithm. The writing was clear and easy to follow. I especially liked how the authors tried to exploit possible pitfalls of their experimental design.

Minor comments and questions:
- Although the problem of interest is non-trivial and important, the proposed algorithm can be seen as just a naive combination of clustering and meta-learning. It would have been great to see some clustering algorithm that was specifically designed for this type of problem. Especially, the proposed CACTUs algorithm relies on sampling without replacement from the clustered dataset in order to enforce "balance" of the labels among the generated task. This might be leading to suboptimal results since the popularity of each cluster (i.e., how much it represents the whole dataset) is not considered.

- CACTUs seems to be relying on having random scaling of the k-means algorithm in order to induce diversity on the set of partitions being generated. I am a bit skeptical about the effectiveness of such a method for diversity. If this holds, it would be interesting to see the visualization of such a concept.

- Although only MAML was considered as the meta-learning algorithm, it would have been nice to consider one or more candidates to show that the proposed framework is generalizable. Still, I think the experiment is persuasive enough to expect that the algorithm would work well at practice.

- Would there be a trivial generalization of the algorithm to semi-supervised learning?

-------

I am satisfied with the author's response and changes they made to the text. I still think the paper brings significant contributions to the area, by showing that even generating the pseudo-tasks via unsupervised clustering method allows the meta-learning to happen.

---

> ### Author Response · Authors · 2018-11-08
> **Thank you for your insightful comments and feedback!**
>
> “Although only MAML was considered as the meta-learning algorithm, it would have been nice to consider one or more candidates to show that the proposed framework is generalizable. Still, I think the experiment is persuasive enough to expect that the algorithm would working well at practice.”
> To address your suggestion, we have updated the paper to add results (in Tables 1, 2, and 3) obtained with Prototypical Networks [1] as the meta-learner instead of MAML. We find that the improvement of CACTUs over the comparison methods still generally holds, with a few exceptions. We hypothesize the exceptions are due to a dependence of ProtoNets performance on matching train shot with test shot, i.e. on providing the meta-learner with tasks that have supervision commensurate to that expected in held-out tasks. We elaborate on this in the updated paper (“Benefit of Meta-Learning” in Section 4.2).
>
> “Although the problem of interest is non-trivial and important, the proposed algorithm can be seen as just a naive combination of clustering and meta-learning. It would have been great to see some clustering algorithm that was specifically designed for this type of problem.”
> The reviewer is correct in that some more sophisticated clustering methods may be better-suited for our method. We found that this simple procedure (with hyperparameter k fixed) worked surprisingly well across datasets and task structures, and did not see a need to make the method more complex.
>
> “Especially, the proposed CACTUs algorithm relies on sampling without replacement from the clustered dataset in order to enforce "balance" of the labels among the generated task. This might be leading to suboptimal results since the popularity of each cluster (i.e., how much it represents the whole dataset) is not considered.”
> If we view k-means as the hard limit of a mixture of Gaussians and decompose the joint embedding-cluster distribution p(z,c)=p(c)p(z|c), one way the reviewer’s proposal can be realized is sampling from p(c). However, as we mention in Section 5, the datasets we consider (Omniglot and miniImageNet) are evenly balanced amongst classes, and we fear that comparing between sampling from U(c) and p(c) on these datasets may be misleading, as in general datasets can be heavily imbalanced. We leave a thorough evaluation of the question of how to better sample clusters for tasks to future work, but there are a couple of hints we can already think about. First, consider a toy imbalanced dataset for which, after clustering, there is one heavily populated cluster and four small ones. Because we have no guarantees about the meta-test distribution, it is “safer” for the meta-learner to learn to distinguish between all clusters equally well than to have the popular cluster dominate the meta-training task distribution. Second, prior work [2] also considers a similar question (“Trivial parametrization”, page 6), and concludes that uniform sampling over clusters is more suitable.
>
> “CACTUs seems to be relying on having random scaling of the k-means algorithm in order to induce diversity on the set of partitions being generated. I am a bit skeptical about the effectiveness of such a method for diversity. If this holds, it would be interesting to see the visualization of such a concept.”
> We found that this diversity was helpful but not critical for our method to perform well: compare the P=1 and P={50,100} entries in the tables of Appendix F. Other mechanisms for encouraging task diversity would be an interesting direction for future work, and we welcome any suggestions on this front!
>
> “Would there be a trivial generalization of the algorithm to semi-supervised learning?”
> For the scenario in which some labeled data, not necessarily from the same classes as in tasks from meta-test time, is available during meta-training, there are indeed some obvious extensions. One can: i) have some of the tasks be generated only from the labeled data and others from CACTUs, ii) encourage the calculation of partitions to respect the labeled data, and/or iii) use the labeled data as meta-validation to do early stopping and hyperparameter tuning. We leave this for future work.
>
> References
> [1] Snell et al. NIPS 2017, https://papers.nips.cc/paper/6996-prototypical-networks-for-few-shot-learning
> [2] Caron et al. ECCV 2018, https://arxiv.org/abs/1807.05520

---

### Official Review · AnonReviewer1 · 2018-11-06
**Interesting approach but still not finished**

**Rating:** 6
**Confidence:** 3

**Review:**

The paper proposes to employ metalearning techniques for unsupervised tasks. The authors construct tasks in an automatic way from unlabeled data and run meta-learning over the constructed tasks.

Although the paper presents a novel approach and the experiments included in the work show promising results, in my opinion, the paper is still not mature. There are some importants problems:
* The motivation of the paper is weak. The authors include the problem statement as well as the definitions used in the paper without knowing what is the goal of the proposed algorithm. A clear example of a real problem where the proposed framework could be applied is necessary to motivate the work.
* The paper is difficult to read and follow. The paper is composed by a set of parts without many links. This makes difficult to read the paper to not very experienced readers. A running example could be useful to increase the readability of the work. In my opinion, the paper contains too much material for the length of the conference. In fact, some important information has been moved to the appendices.
*Experimental section is specially hard to follow. The authors want to solve too many questions in a short space. Comparisons with other related papers should be included.

---

> ### Author Response · Authors · 2018-11-06
> **Thank you for the feedback. Can you elaborate on your suggestions?**
>
> Thank you for your comments. Our evaluation tests on few-shot Omniglot, miniImageNet, and CelebA classification datasets, which are a real-world few-shot image classification task proposed by [1,2,3] respectively, and evaluated in virtually all few-shot classification papers since 2016: [1,2,3,4,5,6,7,8,9]. We can of course evaluate our method on other problems as well, but the current tasks are real-world image datasets and problems that have been studied extensively in the literature, for which our method achieves excellent results. Are there particular additional datasets that the reviewer would prefer a comparison to? Or anything else we can do to address the concern about the motivation?
>
> We would be happy to revise the problem statement and writing as per the reviewer's suggestions, though we would appreciate more specific pointers about what in particular is difficult to follow. The problem statement is quite simple: we aim to propose an algorithm whereby meta-learning can be used to acquire an efficient few-shot learning procedure without any hand-specified labels during meta-training. This problem is important for two reasons: (1) meta-learning currently relies on large labeled datasets, and in practice, the burden of obtaining such labeled datasets is a major obstacle to widespread use of meta-learning for few-shot classification, and (2) state-of-the-art unsupervised learning methods often neglect downstream use-cases, such as few-shot classification, leaving substantial room for improvement. Our work proposed a way to begin addressing these challenges, and compares extensively to four prior papers [10,11,12,13] and several ablations. Beyond updating the problem statement, are there important comparisons that we missed?
>
> [1] Santoro et al. ICML 2016, http://proceedings.mlr.press/v48/santoro16.pdf
> [2] Ravi & Larochelle ICLR 2017, https://openreview.net/pdf?id=rJY0-Kcll
> [3] Finn et al NIPS 2018, https://arxiv.org/abs/1806.02817
> [4] Vinyals et al. NIPS 2016, https://papers.nips.cc/paper/6385-matching-networks-for-one-shot-learning
> [5] Munkhdalai et al. ICML 2017, https://arxiv.org/abs/1703.00837
> [6] Finn et al. ICML 2017, https://arxiv.org/abs/1703.03400
> [7] Snell et al NIPS 2017, https://papers.nips.cc/paper/6996-prototypical-networks-for-few-shot-learning
> [8] Oreshkin et al. NIPS 2018,
> https://arxiv.org/abs/1805.10123
> [9] Yoon et al. NIPS 2018, https://arxiv.org/abs/1806.03836
> [10] Donahue et al. ICLR 2017, https://arxiv.org/abs/1605.09782
> [11] Caron et al. ECCV 2018, https://arxiv.org/abs/1807.05520
> [12] Berthelot et al. arXiv 2018, https://arxiv.org/pdf/1807.07543
> [13] Chen et al. NIPS 2016, https://arxiv.org/abs/1606.03657

---

### Official Review · AnonReviewer5 · 2018-11-26
**nice and simple idea with well carried and thorough empirical experiments**

**Rating:** 7
**Confidence:** 4

**Review:**

summary
The goal of meta-learning is to train a model on a variety of learning tasks, such that it can solve new learning tasks using only a small number of training samples. SoTA meta-learning frameworks (MAML and ProtoNet) typically require rather large labeled datasets and hand-specified task distributions to define a sequence of tasks on which the algorithms are trained on. This paper proposes to unsupervised generate the sequence of tasks using multiple partitions as pseudo labels via k-means and other clustering variants on the embedding space. Empirical experiments show the benefit of the meta-learning on the M-way K-shot image classification tasks.  Also, “sampling a partition from U(P)” on page 4, the U(P) notation seems not defined.

Evaluation
- The writing and presentation of the paper are in general well carried, except some part seems a little unclear, taking me quite a while to understand. For example,  in the “task generation for meta-learning” paragraph on page 3, the definition of task-specific labels (l_n) is puzzling to me at first glance.

- The proposed task construction in an unsupervised manner for the meta-learning framework is indeed simple and novel.

- The empirical experiments are thorough and well-conducted with good justifications. The benefit of unsupervised meta-learning compared to simply supervised learning on the few-shot downstream tasks is shown in Table 1 and 2; Different embedding techniques have also been studied; the results of Oracle upper bound are also presented; task construction ablation is also shown.

- Unsupervised meta-learning consists of multiple components such as learning embedding space, clustering methods, and various choices within the meta-learning frameworks. This together consumes a lot of hyper-parameters and the choice can somehow seem heuristic.

Conclusion
- In general, I like this paper especially the empirical analysis section. Therefore, I vote for accepting this paper.

---

> ### Author Response · Authors · 2018-11-27
> **Thank you for your review. We appreciate your thorough and accurate summary as well as your feedback.**
>
> We have addressed your comments on presentation by specifying U(P) as the uniform distribution and revising the explanation in “task generation for meta-learning”.
>
> We agree that the entire pipeline consists of several hyperparameters, which we chose and fixed based on prior work and heuristics (Section 4.1, Appendix E). We found it to be straightforward to select these parameter values, suggesting that the algorithm is not particularly sensitive to their values.

---

### Author Response · Authors · 2018-11-13
**Paper updated to address reviewer feedback**

We have updated the paper with the following changes to address reviewer comments:
- combined sections 2.1 and 2.3, and sections 2.2 and 2.4 (R2)
- reduced redundancy in the exposition (R2)
- added more mathematical details to section 2 (R2)
- added comparison to clustering on pixels (R2)
- added further discussion of limitations of our method in the discussion (R2)
- provided more motivation and justification for our approach in section 2.2 (R1, R2)
- improved the clarity of the problem statement and its motivation in sections 1 and 2.1 (R1, R2)
- emphasized throughout the text that the downstream tasks we evaluate on at meta-test time are standard benchmark few-shot learning tasks (R1, R2)
- added a brief discussion on sampling clusters in section 2.2 (R3)
- added a set of experiments based on Prototypical Networks (R3)

We would appreciate it if the reviewers could take a look at our changes and additional results, and let us know if they would like to either revise their rating of the paper, or request additional changes that would alleviate their concerns. Thank you!

---

### Public Comment · (anonymous) · 2018-11-27
**Interesting work and a new few-shot setting**

Unsupervised meta-learning seems interesting and a new setting for few-shot learning in this area.

Also, this might be an interesting direction to follow.
Will the code be released after the decision?

---

> ### Author Response · Authors · 2018-11-30
> **Thank you for your interest.**
>
> Code for producing part of the results has been released, but for anonymity reasons, we will not link to it here. We will update the code and add a link to the code in the paper and here after the review process is complete.

---

### Meta-Review · Area_Chair1 · 2018-12-16
**Interesting idea with thorough empirical evaluation**

**Confidence:** 5
**Recommendation:** Accept (Poster)

**Metareview:**

Reviewers largely agree that the paper proposes a novel and interesting idea for unsupervised learning through meta learning and the empirical evaluation does a convincing job in demonstrating its effectiveness. There were some concerns on clarity/readability of the paper which seem to have been addressed by the authors. I recommend acceptance.